

# Application of the LM-BP neural network approach for landslide risk assessments

Junnan Xiong[1,3,*], Ming Sun[2], Hao Zhang[1], Weiming Cheng[3], Yinghui Yang[1], Mingyuan Sun[1], Yifan Cao[1] and Jiyan Wang[1]

[1]School of Civil Engineering and Architecture, Southwest Petroleum University, Chengdu, 610500, P.R. China
[2]Geodetic Third Team, National Administration of Surveying, Mapping and Geo-information of China, Chengdu, 610100, P.R. China
[3]State Key Laboratory of Resources and Environmental Information System, Institute of Geographic Science and Natural Resources Research, Chinese Academy of Sciences, Beijing 100101, P.R. China

*Correspondence to*: Junnan Xiong (neu_xjn@163.com)

Running Title: Landslide risk zonation in pipeline areas



**Abstract.** Landslide disaster is one of the main risks involved with the operation of long-distance oil and
gas pipelines. Because previously established disaster risk models are too subjective, this paper presents
a quantitative model for regional risk assessment through an analysis of the laws of historical landslide
disasters along oil and gas pipelines. Using the Guangyuan section of the Lanzhou-Chengdu-Chongqing
(LCC) Long-Distance Products Oil Pipeline (82km) in China as a case study, we successively carried out
two independent assessments: a hazard assessment and a vulnerability assessment. We used an entropy
weight method to establish a system for the vulnerability assessment, whereas a Levenberg Marquardt-
Back Propagation (LM-BP) neural network model was used to conduct the hazard assessment. The risk
assessment was carried out on the basis of two assessments. The first, the system of the vulnerability
assessment, considered the pipeline position and the angle between the pipe and the landslide (pipeline
laying environmental factors). We also used an interpolation theory to generate the standard sample
matrix of the LM-BP neural network. Accordingly, a landslide hazard risk zoning map was obtained
based on hazard and vulnerability assessment. The results showed that about 70% of the slopes were in
high-hazard areas with a comparatively high landslide possibility and that the southern section of the oil
pipeline in the study area was in danger. These results can be used as a guide for preventing and reducing
regional hazards, establishing safe routes for both existing and new pipelines and safely operating
pipelines in the Guangyuan section and other segments of the LCC oil pipeline.
**Keywords:** pipeline, landslide, risk, vulnerability, hazard, neural network

## 1. Introduction

By the year 2020, the total mileage of long-distance oil and gas pipelines is expected to exceed 160,000
km in China. This represents a major upsurge in the mileage of multinational long-distance oil and gas
pipelines (Huo, Wang, Cao, Wang, & Bureau, 2016). The rapid development of pipelines is associated
with significant geological hazards, especially landslides, which increasingly threaten the safe operation
of pipelines (Wang et al., 2012; Yun & Kang, 2014; Zheng, Zhang, Liu, & Wu, 2012). Landslide disasters
cause great harm to infrastructure and human life. Moreover, the wide impact area of landslides restricts
the economic development of landslide-prone areas (Ding, Heiser, Hübl, & Fuchs, 2016; Hong, Pradhan,
Xu, & Bui, 2015). A devastating landslide can lead to casualties, property losses, environmental damage
and long-term service disruptions caused by massive oil and gas leakages (G. Li, Zhang, Li, Ke, & Wu,
2016; Zheng et al., 2012). Generally, pipeline failure or destruction caused by landslides is much more
deleterious than the landslides themselves, which makes it important to research the risk assessment of
geological landslide hazards in pipeline areas (Inaudi & Glisic, 2006; Mansour, Morgenstern, & Martin,

46    2011).

Natural disaster risk comprises a combination of natural and social attributes (Atta-Ur-Rahman &
Shaw, 2015). The United Nations Department of Humanitarian Affairs expresses natural disaster risk as
a product of hazards and vulnerabilities (Rafiq & Blaschke, 2012; Sari, Innaqa, & Safrilah, 2017). In
recent years, progress in geographic information systems (GIS) and remote sensing (RS) technologies
have greatly enhanced our ability to evaluate the potential risks that landslides pose to pipelines (Akgun,



Kıncal, & Pradhan, 2012; B. Li & Gao, 2015; Sari et al., 2017). The disaster risk assessment model has
been widely recognized and applied by experts and scholars all over the world. Landslide risk assessment
can take the form of a qualitative (Wu, Tang, & Einstein, 1996), quantitative (Ho, Leroi, & Roberds,
2000) or semi-quantitative assessment (Yingchun Liu, Shi, Lu, Xiao, & Wu, 2015) according to actual
demand. Quantitative methods and models that have been proposed for the assessment can be divided
into methods of statistical analysis (Sari et al., 2017), mathematical models (Akgun et al., 2012) and
machine learning (He & Fu, 2009). However, most of these methods are subjective, which could affect
the accuracy and reasonableness of the evaluation (Fall, Azzam, & Noubactep, 2006; Sarkar & Gupta,
2005). This shortcoming can be overcome through the artificial neural network, especially the mature
Back Propagation (BP) Neural Network that is widely used in function approximation and pattern
recognition (Ke & Li, 2014; P. L. Li, Tian, & Li, 2013; Su & Deng, 2003). The evaluation index system
generally includes disaster characteristics, disaster prevention and pipeline attributes (J. Li, 2010;
Shuiping Li, 2008). The fault tree analysis, fuzzy comprehensive evaluation and the grey theory are used
to evaluate the failure probability of the system through index weight and scoring (Shi, 2011; Ye, Jiang,
Yao, Xia, & Zhao, 2013). In previous studies, pipeline vulnerability evaluation indexes only considered
the pipeline itself, and the relationship between the pipeline and environment was rarely examined (Feng,
Zhang, & Zhang, 2014; Shuiping Li, 2008; Yingchun Liu et al., 2015). In this paper, the interaction
between landslide hazards and the pipeline itself was considered, which improved the quantitative degree
of the evaluation.
Based on the theory of the LM-BP neural network, a standard sample matrix was developed using the
interpolation theory after an analysis of the distribution characteristics of landslides that occurred in the
study area was performed and a regional landslide hazards assessment was completed. Considering the
interaction between landslide disasters and the pipeline itself, the pipeline vulnerability evaluation in the
landslide area was realized using the entropy weight method. This paper established a risk assessment
model and methods for assessing landslide geological hazards of oil pipelines by comprehensively
utilizing GIS and RS technology, which together improved the quantitative degree of the assessment.
**2. Study Area**
The study area was Guangyuan City in the Sichuan province, which was further restricted to the area
from 105 °15 ´to 106 °04 E and 32 °03 to 32 °45 N, straddling 19 townships in five counties from south to
north (Figure 1). The Lanzhou-Chengdu-Chongqing (LCC) Products Oil Pipeline is China's first long-
distance pipeline. It begins in Lanzhou City and runs through the Shanxi and Sichuan provinces (Hao &
Liu, 2008). Our study area covered sloped areas of the range with 5 km on both sides of the Guangyuan
section (82 km) of the oil pipeline. The pipeline within the K558-K642 mileages may be affected by the
slope areas. The Guangyuan section, located in northern Sichuan, is a transitional zone from the basin to
the mountain. It features a terrain of moderate and low mountains, crisscrossed networks of ravines and
a strong fluvial incision. Altitudes in this area range from 328 m to 1505 m. The study area has a
subtropical monsoon climate with four distinctive seasons and annual precipitation measuring about 900
mm to 1,000 mm. Moreover, two large unstable faults (the Central Fault of Longmen Mountain and
Longmen Mountain's Piedmont Fault Zone) make the area geologically unstable and prone to frequent



geological hazards (Shiyuan Li et al., 2012). Guangyuan, through which the pipeline passes, has a high
incidence of landslides, some of which have happened 300 times in the Lizhou and Chaotian districts
(Zhang, Shi, Gan, & Liu, 2011). In this area, landslide geological hazards seriously threaten the safe
operation of the LCC oil pipeline.
**3. Data Sources**
Landslide hazard assessment, pipeline vulnerability assessment and geological hazard risk assessment of
the landslide pipeline were made successively. Digital elevation model (DEM) data with 30 m accuracy
was sourced from the Geospatial Data Cloud (http://www.gscloud.cn/). Precipitation data was
downloaded from the dataset of annual surface observation values in China between the years 1981 to
2010, as published by the China Meteorological Administration (http://data.cma.cn/). This data was
collected from 18 meteorological observatories near and within the study area and interpolated using the
kriging method (at a resolution of 30 m ×30 m). Geological maps and landslide data (historical landslides)
in the study area were obtained from the Sichuan province's geological environmental monitoring station.
RS images (GF-1, multispectral 8 m, resolution 2 m) were provided by the Sichuan Remote Sensing
Center.
The location of the middle line of the pipeline was detected through the direct connection method (i.e.,
the transmitter's output line was directly connected to the metal pipeline) using an RD8000 underground
pipeline detector. Pipeline midline coordinates were measured using total network Real Time Kinematic
technology, and simultaneously, the coordinates of the pipe ancillary facilities (including test piles,
mileage piles and milestones) were acquired. Mileage data obtained through inner pipeline detection was
derived from the China Petroleum Pipeline Company.
**4. Methods**
4.1 Assessment unit
Division precision and the scale of the slope unit (i.e., the basic element for a regional landslide hazard
assessment) were in keeping with the results of the evaluation (Qiu, Niu, ZhaoYannan, & Wu, 2015). A
total of 315 slope units were divided using hydrologic analysis in ArcGIS (v. 10.4) (Fig. 2a). The
irrational unit was artificially identified and modified by comparing GF-1 satellite remote sensing
images. Boundary correction, fragment combination and fissure filling were used for modification.
The object of the pipeline vulnerability assessment in the landslide area was the pipeline. Considering
both previous research and the particulars of the research object, we used a comprehensive
segmentation method based on GIS to divide the pipelines in our study. A total of 180 pipes were
divided in the study area, of which the longest was about 1.7 km, and the shortest was only about 10 m
(Fig. 2b).
4.2 Assessment factors
Based on selection principles of the indicator system and the formation mechanism of landslide
geological hazards, as few indicators as possible were selected to reflect the degree of danger posed by
the landslide as accurately as possible (Avalon Cullen, Al-Suhili, & Khanbilvardi, 2016; Jaiswal, Westen,



& Jetten, 2010; Ray, Dimri, Lakhera, & Sati, 2007). he internal factors in these indicators of the paper
included topography, geological structure, stratigraphic lithology and surface coverage. Similarly, the
external factors included mean annual precipitation (MAP) and the coefficient of the variation of annual
rainfall (CVAR). The correlations between indicators were analyzed using R (v. 3.3.1), and the results
showed a significant correlation between MAP and CVAR (R = 0.99) and between NDWI and NDVI (R
= 0.87). Based on correlation and standard deviation, CVAR and NDWI were eliminated from the
original evaluation system for landslide hazard assessment in the pipeline area (Table 1).
Generally, the evaluation index of pipeline vulnerability as it relates to the relationship between a pipeline
and its surrounding environment is rarely considered. The evaluation indicators in this paper were refined
to include pipeline parameters and the spatial relationship between a pipeline and landslide. The pipelines
in the study area were based in mountainous areas and had been running for many years. All of these
pipelines consisted of high-pressure pipes that were made of steel tubes and had a diameter of 610 mm
for conveying oil. In keeping with the theory of the entropy weight method, these indicators (e.g.,
pressure, materials, diameter and media) were not included in the final evaluation system used to
determine pipeline vulnerability.
4.3 LM-BP neural network Model
The LM algorithm, also known as the damped least square method, has the advantage of local fast
convergence. Its strong global searching ability contributes to the strong extrapolation ability of the
trained network. The BP neural network model, optimized by the LM algorithm, was used to evaluate
the regional landslide hazard in this study. MATLAB 2014 with the *trainlm* training function was used
to implement the LM-BP neural network.
Data from 106 landslide disasters was collected near the research area. Of these landslides, 23 were
within the region of the study area. Most of the landslides located outside the study area were less than
20 km away from the pipeline. Due to comparable environmental conditions, these landslides could still
help us identify the relationship between landslides and environment factors. In light of the frequency
distribution of each evaluation indicator (Fig. 3), the landslide hazard grade corresponding to each
interval of the indicators was divided, and then the hazard degree monotonicity in each interval was
decided. For this study, the landslide hazard grade was divided into four levels: low (I), medium (II),
high (III) and extremely high (IV).
On the basis of the classification criteria of the evaluation indicators used to predict landslide hazard
degree and the functional relationship between the evaluation indicators and landslide probabilities,
standard samples (training samples and test samples) were built using a certain mathematical method.
The training samples and test samples were evaluated using similar construction methods but with
different sample sizes. Finally, the indicator data was normalized, it was entered into the LM-BP neural
network for simulation and 315 slope unit landslide hazard values were output.
4.4 Vulnerability assessment model for pipelines
The vulnerability evaluation model of pipelines in the landslide area was established using the entropy
weight method, which overcame the shortcomings of the traditional weight method that does not consider
the different evaluation indexes and the excessive human influence on the process of evaluation (Gao,





Li, Wang, Li, & Lin, 2017; Pal, 2014). Pipeline defect density was obtained from the pipeline internal
inspection data, which consisted of both mileage data that needed to be converted into three-dimensional
coordinate data and pipeline center line coordinate data obtained through C# programming. In addition,
the main slide direction of the landslide was replaced by the slope direction that was extracted by DEM.
The coordinate azimuth of the pipe section was extracted using the linear vector data of each pipe section,
and the angle between the pipeline and the slope was calculated using the mathematical method. The
calculation process was solved in the VB language on ArcGIS using second development functions.
Finally, the entropy weight of 5 indexes was calculated by programming in MATLAB 2014. The entropy
weight calculation results for pipeline landslide vulnerability assessment are shown in Table 2. Pipeline
vulnerability in landslide area was calculated using the following formula:
$$H_j = \sum_{i=1}^{m} w_i r_{ij} \tag{1}$$
where $H_j$ is the evaluation value of the pipeline section's vulnerability; $w_i$ is the weight of the evaluation
index; and $r_{ij}$ represents the $i$th evaluation index values of $j$th pipe sections.

**5 Results and comparison**

5.1 Regional landslide hazard assessment

The LM-BP neural network was trained and the network was stopped after 182 iterations. An RMSE
value of 9.93e-09 indicated that the goal of precision had been reached. Through the simulation of the
network test, none of the absolute error values of test data (20 groups) were found to be greater than 0.02;
this result aligned with our expectation of the precision of the landslide hazard assessment. The landslide
hazard grade was divided into four levels by using the equal interval method at intervals of 0.25. The
safe section (low hazard) was located in the central part of the study area. The dangerous (high hazard)
section was located north and south (Fig. 4). In the study area, most of the exposed rock was dominated
by shale, which belonged to the easy-slip rock group.
Average altitude ranged from 450 m to 1400 m, and the relative height difference was greater than 80
m, with the slope between 15° and 35°. Based on an overlay analysis of historic landslides within the
study area, and hazard zonation maps, we surmised that the probability of landslides in the study area
was extremely high, and that 87% of the landslides occurred in the medium-, high-, and extremely high-
hazard areas. Among these landslides, three were located in low-hazard areas, which accounted for 13%
of the landslide disaster sites, five occurred in medium-hazard areas (accounting for 21.7 of disaster
sites), seven occurred in high-hazard areas (accounting for 30.4% of sites) and eight occurred in
extremely high-hazard areas (accounting for 34.8% of sites). The evaluation results were found to
accurately reflect the trends and rules of distribution of landslides in the study area. The number and area
of slopes in high-hazard and extremely high-hazard areas accounted for about 70% of the total (Table 3).
The probability of landslide occurrence in the study area was generally high, which was consistent with
the fact that the region was landslide-prone.
5.2 Vulnerability assessment for oil pipeline in landslide area



The equal interval of 0.25 was used to divide the pipeline vulnerability level into four grades to obtain
the pipeline vulnerability zonation of the study area (Fig. 5). The pipeline in the northern part of the study
area was given a low vulnerability grade, while the situation in the south of the region is more serious.
The number, length and percentage of pipeline segments with different grade vulnerabilities are shown
in Table 4. The number and length of pipeline segments in highly vulnerable areas (III) and extremely
vulnerable areas (IV) accounted for about 12% of the total.
5.3 Risk assessment for oil pipeline in landslide area
According to natural disaster risk expressions released by the UN, the definition of risk may be expressed
as the product of landslide hazard in a pipeline area and pipeline vulnerabilities in the landslide area. The
risk degrees were distinguished using the equal interval method, and four grades were generated. Where
the comprehensive risk assessment value was within 0 to 0.0625, the corresponding risk grade was Grade
I; the corresponding risk grades with the values of 0.0625 to 0.25, 0.25 to 0.5625 and 0.5625 to 1.0 were
Grade II, III and IV, respectively. The risk grade of each section of the pipeline within the research area
is shown in Fig. 6.
The number of sections with a high-risk grade was 33, which accounted for 18.33% of all pipeline
sections and represented 16.57% of the total pipeline length of 13.461 km). There were 4 sections with
extremely high-risk grade, which accounted for 2.22% of all sections and represented 3.31% of the total
pipeline length of 2.538 km. The section number and length of pipelines lying in high-risk (III) and
extremely high-risk (IV) areas accounted for 20% of the total pipeline length, and the risk grade of
pipelines inside Qingchuan and Jian'ge County was relatively high.
5.4 Analysis of risk assessment results
Large or huge landslides were common in areas that we categorized as extremely high risk, which we
defined as those that were geologically evolving or had experienced obvious deformations within the last
2 years with still visible cracks. These pipelines were subject to dangers at any time, as the pipelines
within the areas prone to landslides were found to contain many defects or extensive damage. These
areas also posed considerable threats; for example, pipeline ruptures or breaks could lead to leakages or
serious deformations that cause transportation failure. Because these are unacceptable events, risk
prevention and control measures must be taken in a short time. Pipelines with extremely high risk were
mainly distributed in the following areas: (1) Xiasi Village in Xiasi County (Pile No. K628-K630); (2)
Shiweng Village-Maliu Village of Xiasi County (Pile No. K635-K637). This section lay in the south of
the research area, with an altitude of 500 m to 750 m. Here, the slope conditions affected the distribution
of groundwater pore pressure and the physical and mechanical characteristics of the rock and soil in three
areas: vegetation cover, evaporation and slope erosion. Ultimately, these three factors affected slope
stability (Luo & Tan, 2011). Vertical and horizontal ravines have also been seen in this section, with
a relative height difference greater than 100 m and slop between 15 ° to 35 °. Slope degrees with
obvious changes had a great influence on slope stability (Chang & Kim, 2004; Hu, Xu, Wang, Asch, &
Hicher, 2015). The exposed rocks in this area were mainly shale and belonged to the sliding-prone
rock group. Rock type and interlayer structure were found to be important internal indicators that a
landslide could occur (Guzzetti, Cardinali, & Reichenbach, 1996; Xiang et al., 2010; Xin, Chong, &



Dai, 2009). The distance between the fault and the pipeline in the section was about 2 km with a
NDVI of about 0.75 and MAP of about 970 mm. Faulted zones and nearby rock and earth masses
that were destroyed in a geologic event reduced the integrity of a slope, and the faults and important
groundwater channels could also cause deformation and damage of a slope (Yinghui Liu, 2009). The
pipelines in these areas exhibited many defects. Most pipelines passed through the slope in an inclined
or horizontal way, an attribute that typically increased the risk of a landslide occurring.
In high-risk areas, small or moderate landslides commonly occurred in areas that we categorized as
high risk. They were in deformation, or had obvious deformation recently (within 2 years), such as
obvious cracks, subsidence or tympanites on the landslide and even shear. The pipelines in these areas
had defects and were buried at a shallow depth. If a landslide occurred in this pipeline area, it could cause
pipe suspension, floating and damage. It could also contribute to a small to moderate leakage of the
medium. However, damaged pipes can be welded or repaired. Monitoring is critical in high-risk areas.
In our study, the pipeline high-risk area was defined by the following areas: (1) Xiasi Town Xiasi Village-
Shiweng Village (pipe No. K622-K633). (2) Xiasi Town Maliu Village Jinzishan Xiangdasang Village
(pipe No. K635-K642). This area was located in the south of the pipe, which was buried in the study area.
The altitude of the study area was between 450 m and 800 m, the relative elevation difference was over
100m and the slope was between 15 ° and 40 °. Most of the outcrops in this area were quartz sandstone,
which belonged to the easy-sliding rock group. The pipes in this area were about 2.5 km away from faults.
The NDVI was about 0.6 to 0.8, and MAP was about 970 nm. Pipes showed many defects, most of them
either crossing the slope or lying in the center of slope. All of the above factors provided sufficient
conditions for the formation of landslide.
In the medium-risk areas, only small landslides were found to occur, and we observed no sign of
deformation. But through the analysis of geological structure, topography and landform, we found the
area to demonstrate a tendency for developing landslides. The pipes in this risk area exhibited almost no
faults and were buried deep beneath the ground. However, under bad conditions, the landslides in these
areas could also affect the pipes' safety, causing the pipes to become exposed or deformed. These areas
need simple monitoring. For our study, medium-risk areas were defined as follows: (1) Sanlong village
of Dongxihe township-Panlong town Dongsheng village (pipe No. K559-K593). (2) Panlong town
Qinlao village-Wu'ai village (pipe No. K595-K597). (3) Baolun town Laolin'gou village-Xiasi town
Youyu village (pipe No. K599-K630).
In the low-risk areas, landslides didn't occur under ordinary conditions, but they could occur if a strong
earthquake hit or if the area experienced continuous or heavy rain. The pipes in low-risk areas showed
no defects and were buried very deep. They were also located far away from areas affected by landslides.
Therefore, landslides in these areas caused no obvious damage to the pipes, and few threatened the safety
of pipes. However, regular inspection is necessary to ensure that the pipes continue to operate safely. The
pipe low-risk area were defined as follows: (1) Panlong town Dongsheng village-Qinlao village (pipe
No. K591-K597). (2) Baolun town Xiaojia village-Baolun town Laolin'gou village (pipe No. K599-
K608).
Through comprehensive analysis of each risk level area, we compiled a list of pipeline landslide risks
(Table 6). This list describes each landslide risk level in four respects: pipeline risk, landslide hazard,



pipeline vulnerability and risk control measures.
**5 Results and comparison**
The faults inherent to traditional landslide risk assessment include excessive human influence, failure of
pipeline vulnerability assessments to consider the interaction between landslide disaster and pipeline
ontology and the low quantification degree of risk assessment results.
Taking the Guangyuan section (82 km) of the LCC oil and gas pipeline as an example, we used GIS
and RS technology to establish a regional landslide hazard assessment model based on the LM-BP neural
network. We determined that there were 112 and 108 slopes in high-hazard and extremely high-hazard
areas that accounted for 33.18% and 40.46% of the total area of the study area, respectively. Then, we
established the model of pipeline vulnerability evaluation based on the entropy weight method by
combining the pipeline body and the environmental information. The number and length of pipe
segments in the highly vulnerable (III) and extremely vulnerable area (IV) accounted for about 12% of
the total. Finally, based on the hazard assessment and the vulnerability assessment, we completed the
risk assessment and risk division of the oil pipeline, thus forming a geological disaster risk assessment
model and a method for oil pipeline and landslide risk assessment. The risk assessment results
demonstrated that the number and length of high-hazard and extremely high-hazard pipeline segments
represented 20% of the total. Similarly, the pipeline risk within Qingchuan and Jian'ge Counties was
relatively high. Our pipeline landslide risk assessment has laid a foundation for the future study of
pipeline safety management and pipeline failure consequence loss assessment.

**Acknowledgments**
The study has been funded by the Strategic Priority Research Program of Chinese Academy of Sciences
(XDA20030302), IWHR(China Institute of Water Resources and Hydropower Research) National
Mountain Flood Disaster Investigation Project (SHZH-IWHR-57), Southwest Petroleum University Of
Science And Technology Innovation Team Projects (2017CXTD09) and the Study on temporal and
spatial differentiation of historical mountain flood disasters in Fujian province (NDMBD2018003).




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





**List of tables and figures**
**Table 1** Indicators of landslide hazard assessment and pipeline vulnerability assessment
**Table 2** Entropy weight of evaluation index
**Table 3** Number and area of slopes of four hazard grade
**Table 4** Number and distances of pipeline of four vulnerability grade
**Table 5** Number and distances of pipeline of four risk grade
**Table 6** Description of pipeline risk level

**Figure 1** Landslide location map of the study area
**Figure 2** All slope units (a) and pipeline section (b) in the study area
**Figure 3** The frequency distribution of each factor in the landslide location. Maps (a), (b), (c), (d), (e),
(f), (g), and (h) represent the elevation, slope, aspect, height difference, TPC, NVI, MAP, and distance
from the fault, respectively
**Figure 4** Landslide hazard map of study area
**Figure 5** Pipeline vulnerability map of study area
**Figure 6** Pipeline risk map of study area






**Table 1**

| | Factor | Indicators |
|---|---|---|
| Landslide hazard index | Landform | Elevation |
| | | Slope |
| | | Aspect |
| | | Height Difference |
| | | Topographic profile curvature (TPC) |
| | Land cover | NDVI |
| | | NDWI |
| | Geology | Lithology |
| | | Distance from the fault |
| | Precipitation | Mean annual precipitation (MAP) |
| | | Coefficient of variation of annual rainfall (CVAR) |
| Pipeline vulnerability index | Pipe Body | Defect Density |
| | | Depth |
| | | Thickness |
| | | Pressure |
| | | Materials |
| | | Diameter |
| | | Media |
| | Spatial relationship between pipeline and landslide | Position |
| | | Angle |






**Table 2**

|  | Depth | Angle | Defect Density | Thickness | Position |
| --- | --- | --- | --- | --- | --- |
| Weight | 0.010007 | 0.101553 | 0.678851 | 0.154322 | 0.055266 |
| Entropy | 0.997322 | 0.97282 | 0.818308 | 0.958696 | 0.985208 |






**Table 3**

| Landslide hazard | Number of slopes | Percentage | Area (km²) | Percentage |
|---|---|---|---|---|
| Low (I) | 33 | 10.48% | 32.63 | 8.76% |
| Medium (II) | 62 | 19.68% | 65.53 | 17.60% |
| High (III) | 112 | 35.56% | 123.55 | 33.18% |
| Extremely high (IV) | 108 | 34.29% | 150.65 | 40.46% |
| Total | 315 | 100% | 372.36 | 100% |






**Table 4**

| Pipeline vulnerability | Number of pipelines | Percentage | Area (km$^2$) | Percentage |
|---|---|---|---|---|
| Low (I) | 120 | 66.66% | 50.417 | 62.06% |
| Medium (II) | 37 | 20.56% | 20.888 | 25.72% |
| High (III) | 22 | 12.22% | 9.833 | 12.11% |
| Extremely (IV) | 1 | 0.56% | 0.087 | 0.11% |
| Total | 180 | 100% | 81.225 | 100% |








**Table 5**

| Pipeline risk | Number of pipelines | Percentage | Area (km$^2$) | Percentage |
|---|---|---|---|---|
| Low (I) | 37 | 20.56% | 14.469 | 17.81% |
| Medium (II) | 106 | 58.89% | 50.757 | 62.49% |
| High (III) | 33 | 18.33% | 13.461 | 16.57% |
| Extremely (IV) | 4 | 2.22% | 2.538 | 3.13% |
| Total | 180 | 100% | 81.225 | 100% |






**Table 6**

| Pipeline risk | landslides hazard | Vulnerability | Risk | Control measures |
|---|---|---|---|---|
| Low (I) | The landslide won't happen under ordinary conditions, but it will occur when strong earthquake, long continuous rain or extremely heavy rain happened. | The pipes in low risk areas have no any defects and buried very deep. Meanwhile, they are far away from the area affected by landslide. | Landslides have no obvious damage to the pipes, and few threats to pipes' safety. | Regular Inspection |
| Medium (II) | Small landslide mainly occur, and no sign of deformation. But through analyzing geological structure, topography and landform, there is a tendency of landslide. | The pipes in risk areas have almost no faults and buried deep. However, under bad condition, the landslide may also affect the pipes' safety. | The landslide may make the pipes exposed or deformation. | simple monitoring |
| High (III) | Landslides are most in medium-model and little-model, and they are in deformation, or have obvious deformation recently, such as obvious cracks, subsidence or tympanites on the landslide and even shear. | The pipeline has defects, and buried shallow. Once landslides occurred in the pipeline area, pipes' safety will be threatened | The safety of pipeline will be threatened and may suffer from pipe suspension, floating, and damage etc. Therefore it will contribute to a small amount of medium leakage. Fortunately, the pipe can be welded or repaired. | Main monitoring |
| Extremely high (IV) | Large or huge landslide is common in the area with extremely high risk, which is changing or has experienced obvious deformation recently with visible cracks. | The pipelines are subject to dangers at any time as the pipelines within the area prone to landslide have been spotted with many defects or much damage. | There are great threats, for example pipeline rupture or break and may lead to considerable leakage of media or serious deformation even transportation failure. | Prevention and control measures shall be taken in a short time |





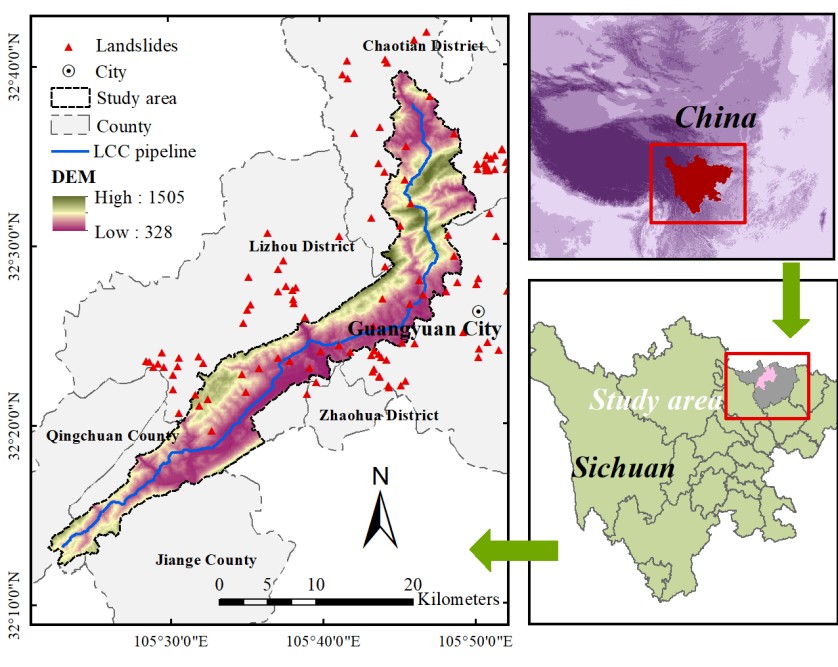


479                                              Figure 1







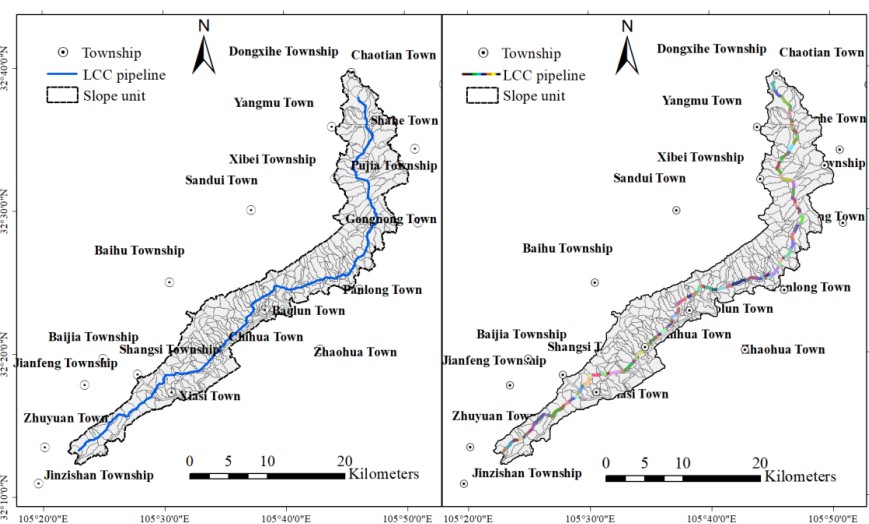


483                                        Figure 2


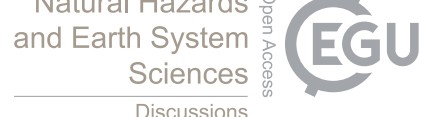

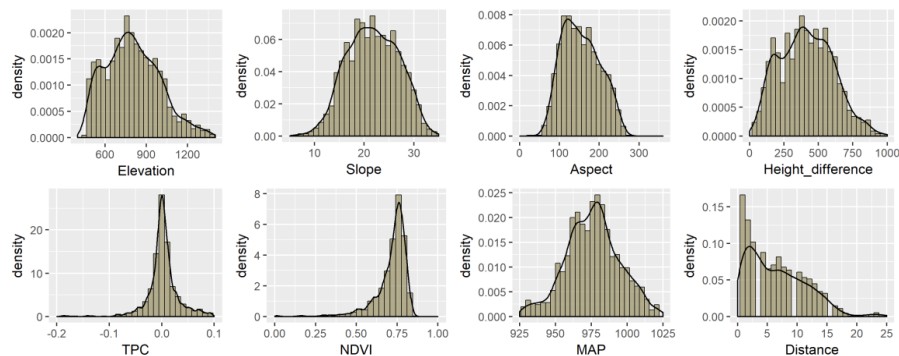


486                                              Figure 3






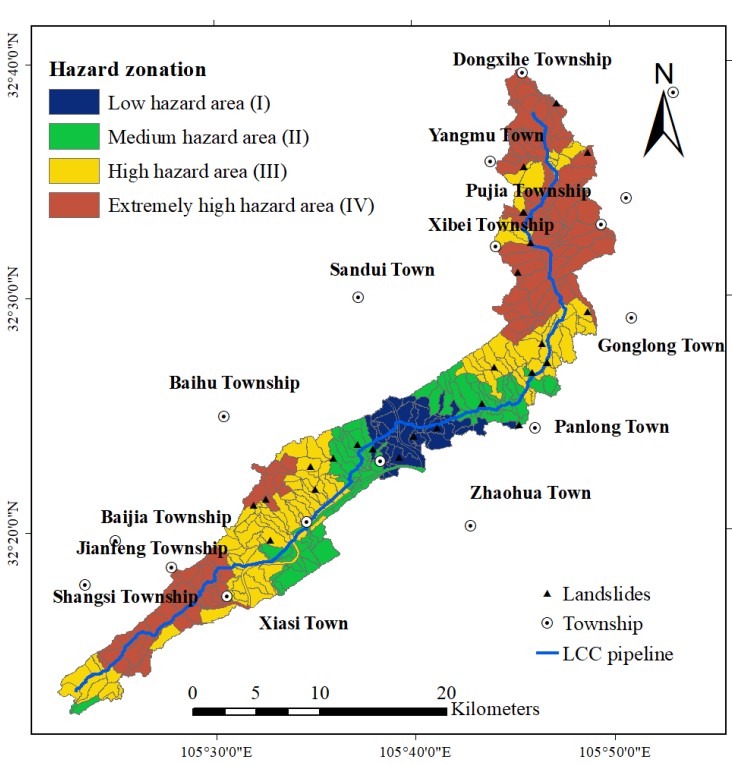


489                                    Figure 4








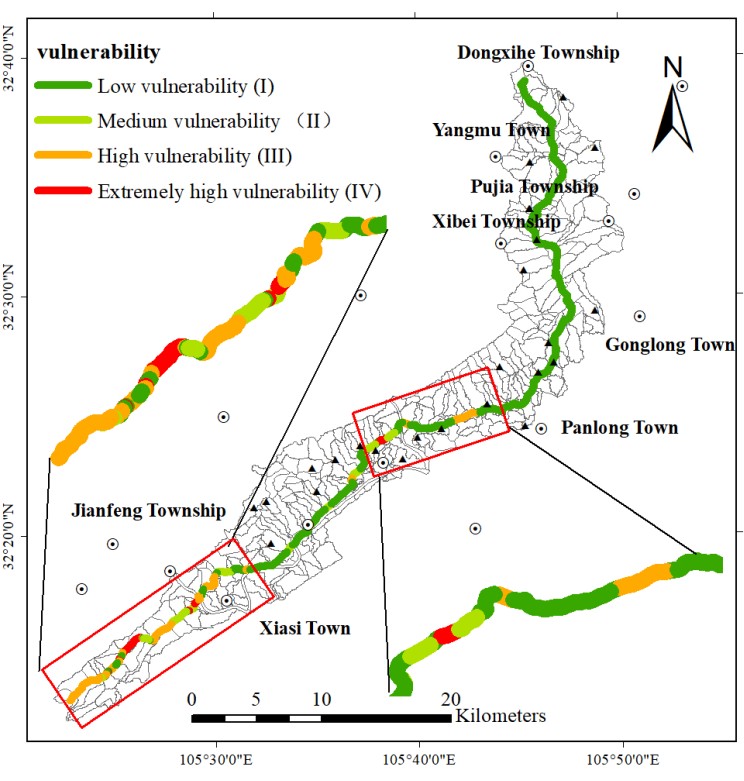


494                                                   Figure 5






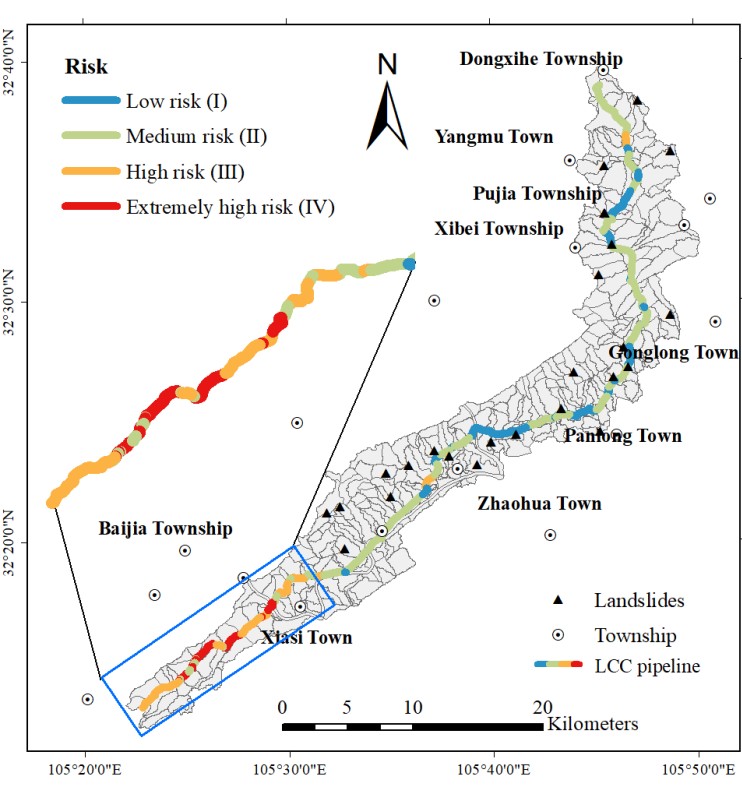


Figure 6







**Appendix 1 Classification of landslide hazard grade corresponding to different intervals**

| Factor | Indicators | Interval | Hazard degree monotonicity | Hazard level |
|---|---|---|---|---|
| Landform | Elevation | [1000 , Highest] | Decreasing | Low hazard(I) |
| | | [Lowest , 600) | Increasing | Medium hazard(II) |
| | | [800 , 1000) | Decreasing | High hazard(III) |
| | | [600 , 700) ∪[700 , 800) | Increasing, Decreasing | Extremely high hazard(IV) |
| | Slope | [60 , 90) | Decreasing | Low hazard(I) |
| | | [0 , 15) | Increasing | Medium hazard(II) |
| | | [30 , 60) | Decreasing | High hazard(III) |
| | | [15 , 20)∪[20 , 30) | Increasing, Decreasing | Extremely high hazard(IV) |
| | Aspect | [0 , 45) ∪[270 , 360) | Increasing, Decreasing | Low hazard(I) |
| | | [225 , 270)∪[45 , 90) | Decreasing, Increasing | Medium hazard(II) |
| | | [90 , 135) ∪[180 , 225) | Increasing, Decreasing | High hazard(III) |
| | | [135 , 157.5) ∪[157.5 , 180) | Increasing, Decreasing | Extremely high hazard(IV) |
| | Height difference | [Lowest , 100) | Increasing | Low hazard(I) |
| | | [900 , Highest] ∪[100 , 200) | Decreasing, Increasing | Medium hazard(II) |
| | | [600 , 900) ∪[200 , 300) | Decreasing, Increasing | High hazard(III) |
| | | [300 , 450)∪[450 , 600) | Increasing, Decreasing | Extremely high hazard(IV) |
| | topographic profile curvature | [Lowest , -0.025) | Increasing | Low hazard(I) |
| | | [0.025 , Highest] | Decreasing | Medium hazard(II) |
| | | [-0.025 , -0.01)∪[0.01 , 0.025) | Increasing, Decreasing | High hazard(III) |
| | | [-0.01 , 0)∪[0 , 0.01) | Increasing, Decreasing | Extremely high hazard(IV) |
| Land cover | NDVI | [-1,0) | Increasing | Low hazard(I) |
| | | [0,0.6)∪[0.9,1] | Increasing, Decreasing | Medium hazard(II) |
| | | [0.6,0.7)∪[0.8,0.9) | Increasing, Decreasing | High hazard(III) |
| | | [0.7,0.75)∪[0.75,0.8) | Increasing, Decreasing | Extremely high hazard(IV) |
| Precipitation | Mean annual precipitation | [1100 , Highest) | Decreasing | Low hazard(I) |
| | | [Lowest , 960) | Increasing | Medium hazard(II) |
| | | [990 , 1100) | Decreasing | High hazard(III) |
| | | [960 ,975)∪[975 , 990) | Increasing, Decreasing | Extremely high hazard(IV) |
| Geology | Distance from the fault | [20, Highest] | Decreasing | Low hazard(I) |
| | | [15 , 20) | Decreasing | Medium hazard(II) |
| | | [5 , 15) | Decreasing | High hazard(III) |
| | | [0 ,5) | Decreasing | Extremely high hazard(IV) |




**Appendix 2 Standard training sample matrix and standard test sample matrix**

| Sample type | ID | Input | | | | | | | | | Output |
| --- | --- | --- | --- | --- | --- | --- | --- | --- | --- | --- | --- |
| | | Aspect | Slope | Elevation | NDVI | MAP | Height Difference | TPC | Distance | Lithology | |
| Training sample | 1 | 0.2 | 89.9 | 438 | -1 | 908.1 | 33 | -0.582 | 25 | 1 | 0 |
| | 50 | 35.2 | 82.8 | 453 | 0 | 912.2 | 79 | -0.456 | 23.47 | 1 | 0.06 |
| | 100 | 297.1 | 75.7 | 469 | 0.88 | 916.3 | 115 | -0.33 | 21.9 | 1 | 0.12 |
| | 150 | 329.3 | 67.6 | 485 | 0.95 | 920.4 | 167 | -0.168 | 20.34 | 1 | 0.19 |
| | 200 | 359.5 | 60 | 499 | 1 | 924.9 | 200 | 0.628 | 18.77 | 1 | 0.25 |
| | 250 | 68.4 | 3.8 | 1293 | 0.73 | 930.4 | 1097 | 0.486 | 17.21 | 2 | 0.31 |
| | 300 | 89.3 | 8.2 | 1206 | 0.65 | 938 | 1039 | 0.326 | 15.64 | 2 | 0.37 |
| | 350 | 246 | 12 | 1102 | 0.56 | 943.6 | 977 | 0.183 | 14.08 | 2 | 0.44 |
| | 400 | 269.3 | 15 | 1002 | 0.5 | 949.8 | 902 | -0.142 | 12.52 | 2 | 0.5 |
| | 450 | 113.4 | 52.9 | 952 | 0.46 | 960.6 | 848 | -0.018 | 10.95 | 3 | 0.56 |
| | 500 | 134.8 | 46.3 | 905 | 0.4 | 972.6 | 757 | -0.012 | 9.39 | 3 | 0.62 |
| Test sample | 1 | 27.2 | 72.3 | 458 | 0.8 | 911.6 | 59 | -0.544 | 25 | 1 | 0 |
| | 2 | 28.5 | 71.6 | 468 | 0.81 | 914.3 | 74 | -0.453 | 23.69 | 1 | 0.06 |
| | 3 | 31.5 | 69.5 | 488 | 0.85 | 915.8 | 86 | -0.381 | 22.37 | 1 | 0.11 |
| | 4 | 37.8 | 66.2 | 490 | 0.86 | 917.1 | 100 | -0.228 | 21.06 | 1 | 0.16 |
| | 5 | 38.6 | 62.1 | 497 | 0.86 | 919.1 | 152 | -0.03 | 19.74 | 1 | 0.22 |
| | 6 | 56.1 | 4.4 | 1141 | 0.7 | 934.2 | 939 | 0.439 | 18.43 | 2 | 0.27 |
| | 7 | 57.3 | 6.6 | 1240 | 0.68 | 939.6 | 941 | 0.429 | 17.11 | 2 | 0.32 |
| | 8 | 65.3 | 9.8 | 1257 | 0.66 | 945.1 | 1124 | 0.413 | 15.79 | 2 | 0.37 |
| | 9 | 68.2 | 11 | 1290 | 0.56 | 948.8 | 1135 | 0.318 | 14.48 | 2 | 0.43 |
| | 10 | 74.7 | 11.9 | 1382 | 0.53 | 949.9 | 1146 | 0.148 | 13.16 | 2 | 0.48 |
| | 11 | 92.4 | 30.4 | 848 | 0.47 | 963.4 | 613 | -0.019 | 11.85 | 3 | 0.53 |
| | 12 | 92.7 | 31.8 | 853 | 0.45 | 970.5 | 683 | -0.016 | 10.53 | 3 | 0.58 |
| | 13 | 101.9 | 44.7 | 900 | 0.45 | 980.5 | 737 | -0.015 | 9.22 | 3 | 0.64 |





| | | | | | | | | | |
|---|---|---|---|---|---|---|---|---|---|
| 14 | 110.1 | 50.9 | 917 | 0.35 | 987 | 817 | -0.015 | 7.9 | 3 | 0.69 |
| 15 | 115.6 | 57.5 | 933 | 0.32 | 994.2 | 835 | -0.015 | 6.58 | 3 | 0.74 |
| 16 | 140.6 | 15.6 | 502 | 0.14 | 1001.5 | 245 | 0.019 | 5.27 | 4 | 0.79 |
| 17 | 155.4 | 20 | 626 | 0.14 | 1002.3 | 256 | 0.008 | 3.95 | 4 | 0.85 |
| 18 | 157.1 | 24.8 | 690 | 0.08 | 1010.6 | 293 | 0.007 | 2.64 | 4 | 0.9 |
| 19 | 177.6 | 27.3 | 765 | 0.06 | 1012.7 | 392 | 0.004 | 1.32 | 4 | 0.95 |
| 20 | 178.3 | 29.6 | 795 | 0.04 | 1022.7 | 446 | 0.001 | 0 | 4 | 1 |




**Appendix 3 Test error of LM-BP neural network**

| Number | Expected value | network output | error |
|---|---|---|---|
| 1 | 0 | 0.0006 | 0.0006 |
| 2 | 0.06 | 0.0548 | -0.0052 |
| 3 | 0.11 | 0.1113 | 0.0013 |
| 4 | 0.16 | 0.1699 | 0.0099 |
| 5 | 0.22 | 0.2302 | 0.0102 |
| 6 | 0.27 | 0.2614 | -0.0086 |
| 7 | 0.32 | 0.315 | -0.005 |
| 8 | 0.37 | 0.3697 | -0.0003 |
| 9 | 0.43 | 0.4266 | -0.0034 |
| 10 | 0.48 | 0.4899 | 0.0099 |
| 11 | 0.53 | 0.5153 | -0.0147 |
| 12 | 0.58 | 0.5765 | -0.0035 |
| 13 | 0.64 | 0.6405 | 0.0005 |
| 14 | 0.69 | 0.701 | 0.011 |
| 15 | 0.74 | 0.7523 | 0.0123 |
| 16 | 0.79 | 0.8094 | 0.0194 |
| 17 | 0.85 | 0.8616 | 0.0116 |
| 18 | 0.9 | 0.9155 | 0.0155 |
| 19 | 0.95 | 0.9675 | 0.0175 |
| 20 | 1 | 1.0173 | 0.0173 |






**Appendix 4 Coordinates of the center line and ancillary facilities of the pipeline**

| Point number | Previous point | Material | Diameter (mm) | Pressure | Depth (m) | Coordinate X | Y | H | elevation |
|---|---|---|---|---|---|---|---|---|---|
| Marker peg | -- | -- | -- | -- | -- | -576.265 | -4357.849 | 503.877 | -- |
| GD1.421 | GD1.420 | Steel | 168 | high | 2.2 | -572.111 | -4352.109 | 504.235 | 502.035 |
| GD1.422 | GD1.421 | Steel | 168 | high | 1.9 | -571.837 | -4336.010 | 503.866 | 501.966 |
| GD1.423 | GD1.422 | Steel | 168 | high | 2.1 | -571.538 | -4319.679 | 503.694 | 501.594 |
| GD1.424 | GD1.423 | Steel | 168 | high | 2.1 | -571.093 | -4308.825 | 503.510 | 501.410 |
| GD1.425 | GD1.424 | Steel | 168 | high | 2.0 | -570.718 | -4288.141 | 503.733 | 501.733 |
| Detective pole K566 | -- | -- | -- | -- | -- | -575.536 | -4284.069 | 503.494 | -- |
| GD1.426 | GD1.425 | Steel | 168 | high | 2.3 | -570.603 | -4275.147 | 503.998 | 501.698 |
| Mileage peg K566+200 | -- | -- | -- | -- | -- | -574.641 | -4258.41 | 503.224 | -- |
| GD1.427 | GD1.426 | Steel | 168 | high | 2.0 | -570.222 | -4258.593 | 503.710 | 501.710 |
| GD1.428 | GD1.427 | Steel | 168 | high | 1.6 | -570.090 | -4247.642 | 503.283 | 501.683 |
| GD1.429 | GD1.428 | Steel | 168 | high | 2.3 | -569.458 | -4216.618 | 502.468 | 500.168 |
| GD1.430 | GD1.429 | Steel | 168 | high | 2.9 | -569.043 | -4208.558 | 504.055 | 501.155 |





**Appendix 5 Internal detection data of pipeline**

| FID | Pipe number | distance(m) | • Feature type | Remarks | Length (mm) | thickness (mm) |
|---|---|---|---|---|---|---|
| 1 | 10 | 6.408 | Pipe segment | Spiral weld | 652 | 11.1 |
| 2 | 20 | 7.060 | Pipe segment | -- | 1178 | -- |
| 3 | 20 | 7.648 | Fixed punctuation point | Valve centerline | -- | -- |
| 4 | 20 | 7.650 | Valve | centerline | -- | -- |
| 5 | 30 | 8.238 | Pipe segment | Spiral weld | 768 | 11.1 |
| 6 | 40 | 9.006 | Pipe segment | -- | 2184 | -- |
| 7 | 40 | 10.100 | Globular tee | centerline | -- | -- |
| 8 | 50 | 11.190 | Pipe segment | Spiral weld | 1700 | 11.1 |
| 9 | 50 | 11.445 | Pit | -- | 548 | 11.1 |
| 10 | 60 | 12.890 | Pipe segment | Straight weld | 2342 | 13.6 |
| 11 | 60 | 12.890 | Wall thickness variation | from 11.1mmto 13.6mm | -- | -- |
| 13 | 70 | 15.232 | Pipe segment | Spiral weld | 1999 | 11.1 |
| 14 | 70 | 15.232 | Wall thickness variation | from 13.6mmto 11.1mm | -- | -- |
| 15 | 80 | 17.231 | Pipe segment | Straight weld | 2352 | 13.4 |
| 16 | 80 | 17.231 | Wall thickness variation | from 11.1mmto 13.4mm | -- | -- |
| 18 | 90 | 19.583 | Pipe segment | Spiral weld | 11557 | 11.1 |
| 19 | 90 | 19.583 | Wall thickness variation | from 13.4mmto 11.1mm | -- | -- |
| 20 | 90 | 28.060 | Attachments | -- | -- | 11.1 |
| 21 | 100 | 31.140 | Pipe segment | -- | 598 | -- |
| 22 | 100 | 31.580 | Flange | centerline | 991 | -- |
| 23 | 110 | 32.131 | Pipe segment | Spiral weld | 11660 | 11.1 |
| 24 | 120 | 43.791 | Pipe segment | Spiral weld | 5536 | 11.1 |
| 25 | 130 | 49.327 | Pipe segment | Straight weld | 2213 | 16.2 |
| 26 | 130 | 49.327 | Wall thickness variation | from 11.1mmto 16.2mm | -- | -- |





| 28 | 140 | 51.540 | Pipe segment | Spiral weld | 5608 | 11.1 |
| 29 | 140 | 51.540 | Wall thickness variation | from 16.2mm to 11.1mm | -- | -- |
| 30 | 150 | 57.148 | Pipe segment | Spiral weld | 9432 | 11.1 |




**Appendix 6 Core Code of Pipeline Defect Point Coordinate Calculating Program**

```
using System;
using System.Collections.Generic;
using System.ComponentModel;
using System.Data;
using System.Drawing;
using System.Linq;
using System.Text;
using System.Threading.Tasks;
using System.Windows.Forms;
using System.IO;
private void button10_Click(object sender, EventArgs e)
{
double x1 = 0, y1 = 0, z1 = 0, x2 = 0, y2 = 0, z2 = 0, d1 = 0, d2 = 0, h1 = 0, h2 = 0;
double l = Convert.ToDouble(textBox9.Text);
double f = 0,nl=Convert.ToDouble(textBox7 .Text );
string[] SplitTxt = textBox2.Text.Split(',');
for (long   i = 0; i < SplitTxt.Length-9; i+=5)
{
d1 = Convert.ToDouble(SplitTxt[i + 1]);
x1 = Convert.ToDouble(SplitTxt[i + 2]);
y1 = Convert.ToDouble(SplitTxt[i + 3]);
z1 = Convert.ToDouble(SplitTxt[i + 4]);
d2 = Convert.ToDouble(SplitTxt[i + 6]);
x2 = Convert.ToDouble(SplitTxt[i + 7]);
y2 = Convert.ToDouble(SplitTxt[i + 8]);
z2 = Convert.ToDouble(SplitTxt[i + 9]);
h1 = z1-d1;
h2 = z2-d2;
556            l += Math.Sqrt((x1-x2)*(x1-x2)+(y1-y2)*(y1-y2)+(h1-h2)*(h1-h2));
}
textBox8.Text =l.ToString();
f = (nl-l)/nl;
ff = f;
textBox5.Text = Convert.ToDouble(f ).ToString("P");
}
private void button9_Click(object sender, EventArgs e)
{
double f1 = ff ;
double l1 = 0;
string zb = "";    string[] SplitTxt = textBox3.Text.Split(',');
for (long i = 0; i < SplitTxt.Length - 1; i += 2)
{
l1 = Convert.ToDouble(SplitTxt[i + 1]);
```



```
l1 += (-ff) * l1;
double x1 = 0, y1 = 0, z1 = 0, x2 = 0, y2 = 0, z2 = 0, d1 = 0, d2 = 0, h1 = 0, h2 = 0, l0=0,l2=0;
double l = Convert.ToDouble(textBox9.Text);
double x = 0, y = 0, h = 0;
string[] SplitTxt1 = textBox2.Text.Split(',');
for (long j = 0; j < SplitTxt1.Length - 9; j += 5)
{
d1 = Convert.ToDouble(SplitTxt1[j    + 1]);
x1 = Convert.ToDouble(SplitTxt1[j    + 2]);
y1 = Convert.ToDouble(SplitTxt1[j    + 3]);
z1 = Convert.ToDouble(SplitTxt1[j    + 4]);
d2 = Convert.ToDouble(SplitTxt1[j    + 6]);
x2 = Convert.ToDouble(SplitTxt1[j    + 7]);
y2 = Convert.ToDouble(SplitTxt1[j    + 8]);
z2 = Convert.ToDouble(SplitTxt1[j    + 9]);
h1 = z1 - d1; h2 = z2 - d2;
l0= Math.Sqrt((x1 - x2) * (x1 - x2) + (y1 - y2) * (y1 - y2) + (h1 - h2) * (h1 - h2));
588               l = l + l0;
if (l - l1 < 0)
{
;
}
else if (l - l1 >0)
{
l2 = l0 - (l - l1);
x = x1 + (x2 - x1) * l2 / l0;
y = y1 + (y2 - y1) * l2 / l0;
598                    h = h1 + (h2 - h1) * l2 / l0;
string xx, yy, hh, v;
v = SplitTxt[i];
xx = Convert.ToDouble(x).ToString();
yy = Convert.ToDouble(y).ToString();
hh = Convert.ToDouble(h).ToString();
zb   +=v + ","+ xx   + "," + yy + "," + hh +",\n";
break;
}
}
}
textBox6.Text = zb;
}
```





611                    **Appendix 7 Pipeline Landslide Risk Assessment Results**

| Fid | Start | Terminus | Hazard | Hazard level | Vulnerability | Vulnerability level | Risk | Risk level |
|---|---|---|---|---|---|---|---|---|
| 1 | K558 | K559+446 | 0.874 | IV | 0.168 | I | 0.147 | II |
| 2 | K559+446 | K563+718 | 0.874 | IV | 0.178 | I | 0.156 | II |
| 3 | K563+718 | K564+883 | 0.932 | IV | 0.143 | I | 0.133 | II |
| 4 | K564+883 | K566+90 | 0.943 | IV | 0.149 | I | 0.141 | II |
| 5 | K566+90 | K567+117 | 0.943 | IV | 0.280 | II | 0.264 | III |
| 6 | K567+117 | K567+224 | 0.766 | IV | 0.095 | I | 0.073 | I |
| 7 | K567+224 | K567+384 | 0.729 | III | 0.117 | I | 0.085 | II |
| 8 | K567+384 | K567+674 | 0.729 | III | 0.079 | I | 0.058 | I |
| 9 | K567+674 | K567+782 | 0.729 | III | 0.141 | I | 0.103 | II |
| 10 | K567+782 | K567+846 | 0.729 | III | 0.066 | I | 0.048 | I |
| 11 | K567+846 | K567+904 | 0.729 | III | 0.097 | I | 0.071 | I |
| 12 | K568+904 | K568+197 | 0.722 | III | 0.154 | I | 0.111 | II |
| 13 | K568+197 | K568+430 | 0.763 | IV | 0.144 | I | 0.110 | II |
| 14 | K569+430 | K569+419 | 0.739 | III | 0.186 | I | 0.137 | II |
| 15 | K569+419 | K569+443 | 0.739 | III | 0.141 | I | 0.104 | II |
| 16 | K569+443 | K569+467 | 0.739 | III | 0.107 | I | 0.079 | II |
| 17 | K569+467 | K569+578 | 0.739 | III | 0.121 | I | 0.089 | II |
| 18 | K569+578 | K569+920 | 0.739 | III | 0.107 | I | 0.079 | II |
| 19 | K571+920 | K571+123 | 0.736 | III | 0.127 | I | 0.093 | II |
| 20 | K571+123 | K571+982 | 0.799 | IV | 0.109 | I | 0.087 | II |
| 21 | K572+982 | K572+729 | 0.753 | IV | 0.090 | I | 0.068 | I |
| 22 | K573+729 | K573+548 | 0.802 | IV | 0.094 | I | 0.075 | I |
| 23 | K574+548 | K574+249 | 0.805 | IV | 0.084 | I | 0.068 | I |
| 24 | K574+249 | K574+525 | 0.805 | IV | 0.150 | I | 0.121 | II |
| 25 | K575+525 | K575+538 | 0.805 | IV | 0.115 | I | 0.093 | II |
| 26 | K575+538 | K575+600 | 0.805 | IV | 0.157 | I | 0.126 | II |
| 27 | K576+600 | K576+737 | 0.816 | IV | 0.108 | I | 0.088 | II |
| 28 | K577+737 | K577+120 | 0.889 | IV | 0.089 | I | 0.079 | I |
| 29 | K577+120 | K577+146 | 0.889 | IV | 0.094 | I | 0.084 | I |
| 30 | K577+146 | K577+187 | 0.889 | IV | 0.169 | I | 0.150 | II |
| 31 | K578+187 | K578+571 | 0.889 | IV | 0.118 | I | 0.105 | II |
| 32 | K578+571 | K578+608 | 0.889 | IV | 0.095 | I | 0.084 | I |
| 33 | K579+608 | K579+624 | 0.853 | IV | 0.133 | I | 0.113 | II |
| 34 | K580+624 | K580+582 | 0.871 | IV | 0.156 | I | 0.136 | II |
| 35 | K581+582 | K581+43 | 0.871 | IV | 0.097 | I | 0.084 | I |
| 36 | K581+43 | K581+273 | 0.871 | IV | 0.143 | I | 0.125 | II |
| 37 | K581+273 | K581+536 | 0.880 | IV | 0.125 | I | 0.110 | II |
| 38 | K581+536 | K581+659 | 0.872 | IV | 0.154 | I | 0.134 | II |
| 39 | K582+659 | K582+263 | 0.830 | IV | 0.152 | I | 0.126 | II |
| 40 | K582+263 | K582+437 | 0.830 | IV | 0.116 | I | 0.096 | II |
| 41 | K583+437 | K583+512 | 0.830 | IV | 0.152 | I | 0.126 | II |
| 42 | K583+512 | K583+693 | 0.798 | IV | 0.105 | I | 0.084 | II |
| 43 | K583+693 | K583+720 | 0.740 | III | 0.113 | I | 0.084 | II |
| 44 | K585+720 | K585+55 | 0.740 | III | 0.178 | I | 0.132 | III |
| 45 | K585+55 | K585+101 | 0.668 | III | 0.196 | I | 0.131 | II |
| 46 | K585+101 | K585+370 | 0.668 | III | 0.178 | I | 0.119 | II |
| 47 | K585+370 | K585+634 | 0.696 | III | 0.190 | I | 0.132 | II |
| 48 | K585+634 | K585+734 | 0.668 | III | 0.116 | I | 0.077 | II |



| | | | | | | | | |
|---|---|---|---|---|---|---|---|---|
| 49 | K585+734 | K585+908 | 0.627 | III | 0.198 | I | 0.124 | II |
| 50 | K585+908 | K585+949 | 0.627 | III | 0.168 | I | 0.105 | II |
| 51 | K586+949 | K586+782 | 0.627 | III | 0.173 | I | 0.108 | II |
| 52 | K586+782 | K586+805 | 0.627 | III | 0.117 | I | 0.073 | II |
| 53 | K587+805 | K587+364 | 0.627 | III | 0.171 | I | 0.107 | II |
| 54 | K587+364 | K587+498 | 0.618 | III | 0.078 | I | 0.048 | I |
| 55 | K587+498 | K587+794 | 0.618 | III | 0.107 | I | 0.066 | I |
| 56 | K589+794 | K589+251 | 0.618 | III | 0.102 | I | 0.063 | I |
| 57 | K590+251 | K590+757 | 0.618 | III | 0.172 | I | 0.106 | II |
| 58 | K590+757 | K590+780 | 0.556 | III | 0.153 | I | 0.085 | II |
| 59 | K590+780 | K590+812 | 0.556 | III | 0.123 | I | 0.068 | II |
| 60 | K591+812 | K591+500 | 0.555 | III | 0.135 | I | 0.075 | II |
| 61 | K591+500 | K591+946 | 0.555 | III | 0.087 | I | 0.048 | I |
| 62 | K592+946 | K592+259 | 0.555 | III | 0.107 | I | 0.059 | I |
| 63 | K593+259 | K593+631 | 0.517 | III | 0.152 | I | 0.079 | II |
| 64 | K593+631 | K593+912 | 0.374 | II | 0.153 | I | 0.057 | II |
| 65 | K594+912 | K594+993 | 0.374 | II | 0.150 | I | 0.056 | II |
| 66 | K595+993 | K595+203 | 0.374 | II | 0.076 | I | 0.028 | I |
| 67 | K595+203 | K595+261 | 0.359 | II | 0.114 | I | 0.041 | I |
| 68 | K595+261 | K595+383 | 0.359 | II | 0.099 | I | 0.036 | I |
| 69 | K596+383 | K596+383 | 0.412 | II | 0.278 | II | 0.115 | II |
| 70 | K596+383 | K596+429 | 0.412 | II | 0.107 | I | 0.044 | I |
| 71 | K597+429 | K597+62 | 0.359 | II | 0.121 | I | 0.043 | I |
| 72 | K597+62 | K597+200 | 0.412 | II | 0.158 | I | 0.065 | II |
| 73 | K597+200 | K597+345 | 0.412 | II | 0.133 | I | 0.055 | I |
| 74 | K597+345 | K597+680 | 0.412 | II | 0.273 | II | 0.112 | II |
| 75 | K599+680 | K599+376 | 0.321 | II | 0.461 | II | 0.148 | II |
| 76 | K599+376 | K599+693 | 0.211 | I | 0.105 | I | 0.022 | I |
| 77 | K600+693 | K600+188 | 0.211 | I | 0.179 | I | 0.038 | I |
| 78 | K600+188 | K600+353 | 0.106 | I | 0.172 | I | 0.018 | I |
| 79 | K601+353 | K601+369 | 0.106 | I | 0.264 | II | 0.028 | I |
| 80 | K602+369 | K602+495 | 0.099 | I | 0.190 | I | 0.019 | I |
| 81 | K603+495 | K603+131 | 0.067 | I | 0.436 | II | 0.029 | I |
| 82 | K603+131 | K603+551 | 0.099 | I | 0.144 | I | 0.014 | I |
| 83 | K604+551 | K604+321 | 0.104 | I | 0.253 | II | 0.026 | I |
| 84 | K604+321 | K604+976 | 0.099 | I | 0.102 | I | 0.010 | I |
| 85 | K605+976 | K605+735 | 0.178 | I | 0.372 | II | 0.066 | II |
| 86 | K606+735 | K606+368 | 0.236 | I | 0.637 | III | 0.150 | II |
| 87 | K606+368 | K606+838 | 0.236 | I | 0.127 | I | 0.030 | I |
| 88 | K607+838 | K607+596 | 0.323 | II | 0.407 | II | 0.131 | II |
| 89 | K608+596 | K608+20 | 0.323 | II | 0.163 | I | 0.053 | II |
| 90 | K608+20 | K608+287 | 0.323 | II | 0.145 | I | 0.047 | I |
| 91 | K608+287 | K608+546 | 0.346 | II | 0.084 | I | 0.029 | I |
| 92 | K608+546 | K608+583 | 0.406 | II | 0.215 | I | 0.087 | II |
| 93 | K608+583 | K608+835 | 0.406 | II | 0.291 | II | 0.118 | II |
| 94 | K609+835 | K609+565 | 0.442 | II | 0.279 | II | 0.123 | II |
| 95 | K610+565 | K610+564 | 0.442 | II | 0.403 | II | 0.178 | II |
| 96 | K610+564 | K610+945 | 0.442 | II | 0.453 | II | 0.200 | II |
| 97 | K611+945 | K611+89 | 0.482 | II | 0.117 | I | 0.056 | I |
| 98 | K611+89 | K611+691 | 0.501 | III | 0.138 | I | 0.069 | II |
| 99 | K612+691 | K612+413 | 0.501 | III | 0.175 | I | 0.088 | II |



| | | | | | | | | |
|---|---|---|---|---|---|---|---|---|
| 100 | K613+413 | K613+269 | 0.501 | III | 0.163 | I | 0.082 | II |
| 101 | K613+269 | K613+442 | 0.502 | III | 0.166 | I | 0.083 | II |
| 102 | K614+442 | K614+83 | 0.502 | III | 0.354 | II | 0.178 | II |
| 103 | K614+83 | K614+980 | 0.502 | III | 0.263 | II | 0.132 | II |
| 104 | K615+980 | K615+218 | 0.601 | III | 0.153 | I | 0.092 | II |
| 105 | K615+218 | K615+388 | 0.601 | III | 0.143 | I | 0.086 | II |
| 106 | K616+388 | K616+87 | 0.635 | III | 0.126 | I | 0.080 | II |
| 107 | K616+87 | K616+300 | 0.556 | III | 0.144 | I | 0.080 | II |
| 108 | K616+300 | K616+460 | 0.505 | III | 0.269 | II | 0.136 | II |
| 109 | K617+460 | K617+715 | 0.505 | III | 0.172 | I | 0.087 | II |
| 110 | K617+715 | K617+827 | 0.505 | III | 0.255 | II | 0.129 | II |
| 111 | K618+827 | K618+28 | 0.556 | III | 0.170 | I | 0.095 | II |
| 112 | K618+28 | K618+687 | 0.556 | III | 0.313 | II | 0.174 | II |
| 113 | K620+687 | K620+78 | 0.556 | III | 0.188 | I | 0.105 | II |
| 114 | K620+78 | K620+298 | 0.425 | II | 0.196 | I | 0.083 | II |
| 115 | K621+298 | K621+509 | 0.576 | III | 0.223 | I | 0.128 | II |
| 116 | K621+509 | K621+611 | 0.425 | II | 0.107 | I | 0.045 | I |
| 117 | K622+611 | K622+10 | 0.425 | II | 0.262 | II | 0.111 | II |
| 118 | K622+10 | K622+86 | 0.425 | II | 0.122 | I | 0.052 | I |
| 119 | K622+86 | K622+539 | 0.693 | III | 0.178 | I | 0.123 | II |
| 120 | K622+539 | K622+897 | 0.634 | III | 0.549 | III | 0.348 | III |
| 121 | K623+897 | K623+36 | 0.634 | III | 0.535 | III | 0.339 | III |
| 122 | K623+36 | K623+794 | 0.693 | III | 0.145 | I | 0.100 | II |
| 123 | K624+794 | K624+866 | 0.693 | III | 0.310 | II | 0.215 | II |
| 124 | K625+866 | K625+242 | 0.796 | IV | 0.137 | I | 0.109 | II |
| 125 | K627+242 | K627+60 | 0.859 | IV | 0.452 | II | 0.388 | III |
| 126 | K627+60 | K627+162 | 0.859 | IV | 0.193 | I | 0.166 | II |
| 127 | K627+162 | K627+313 | 0.859 | IV | 0.166 | I | 0.143 | II |
| 128 | K627+313 | K627+700 | 0.783 | IV | 0.167 | I | 0.131 | II |
| 129 | K628+700 | K628+146 | 0.908 | IV | 0.501 | III | 0.455 | III |
| 130 | K628+146 | K628+196 | 0.908 | IV | 0.139 | I | 0.126 | II |
| 131 | K628+196 | K628+610 | 0.908 | IV | 0.631 | III | 0.573 | IV |
| 132 | K629+610 | K629+355 | 0.787 | IV | 0.369 | II | 0.290 | III |
| 133 | K629+355 | K629+525 | 0.787 | IV | 0.729 | III | 0.574 | IV |
| 134 | K629+525 | K629+570 | 0.787 | IV | 0.252 | II | 0.198 | II |
| 135 | K629+570 | K629+620 | 0.787 | IV | 0.465 | II | 0.366 | III |
| 136 | K630+620 | K630+348 | 0.787 | IV | 0.286 | II | 0.225 | II |
| 137 | K630+348 | K630+956 | 0.892 | IV | 0.389 | II | 0.347 | III |
| 138 | K631+956 | K631+116 | 0.886 | IV | 0.423 | II | 0.375 | III |
| 139 | K631+116 | K631+528 | 0.805 | IV | 0.513 | III | 0.413 | III |
| 140 | K633+528 | K633+435 | 0.805 | IV | 0.568 | III | 0.457 | III |
| 141 | K635+435 | K635+302 | 0.933 | IV | 0.625 | III | 0.583 | IV |
| 142 | K635+302 | K635+326 | 0.884 | IV | 0.611 | III | 0.540 | III |
| 143 | K635+326 | K635+359 | 0.884 | IV | 0.441 | II | 0.390 | III |
| 144 | K635+359 | K635+368 | 0.884 | IV | 0.194 | I | 0.171 | II |
| 145 | K635+368 | K635+530 | 0.884 | IV | 0.374 | II | 0.331 | III |
| 146 | K635+530 | K635+604 | 0.884 | IV | 0.307 | II | 0.271 | III |
| 147 | K635+604 | K635+850 | 0.805 | IV | 0.377 | II | 0.303 | III |
| 148 | K635+850 | K635+943 | 0.805 | IV | 0.234 | I | 0.188 | II |
| 149 | K635+943 | K635+972 | 0.805 | IV | 0.139 | I | 0.112 | II |
| 150 | K635+972 | K635+974 | 0.805 | IV | 0.121 | I | 0.097 | II |



| 151 | K635+974 | K635+990 | 0.805 | IV | 0.138 | I | 0.111 | II |
| 152 | K636+990 | K636+152 | 0.933 | IV | 0.598 | III | 0.558 | III |
| 153 | K636+152 | K636+159 | 0.933 | IV | 0.157 | I | 0.146 | II |
| 154 | K636+159 | K636+320 | 0.884 | IV | 0.579 | III | 0.512 | III |
| 155 | K636+320 | K636+427 | 0.884 | IV | 0.166 | I | 0.147 | II |
| 156 | K636+427 | K636+517 | 0.884 | IV | 0.124 | I | 0.110 | II |
| 157 | K636+517 | K636+806 | 0.834 | IV | 0.663 | III | 0.553 | III |
| 158 | K636+806 | K636+893 | 0.834 | IV | 0.794 | IV | 0.662 | IV |
| 159 | K637+893 | K637+57 | 0.834 | IV | 0.519 | III | 0.433 | III |
| 160 | K637+57 | K637+109 | 0.834 | IV | 0.542 | III | 0.452 | III |
| 161 | K637+109 | K637+181 | 0.834 | IV | 0.111 | I | 0.093 | II |
| 162 | K637+181 | K637+332 | 0.834 | IV | 0.127 | I | 0.106 | II |
| 163 | K638+332 | K638+87 | 0.834 | IV | 0.608 | III | 0.507 | III |
| 164 | K638+87 | K638+140 | 0.834 | IV | 0.157 | I | 0.131 | II |
| 165 | K638+140 | K638+193 | 0.767 | IV | 0.682 | III | 0.523 | III |
| 166 | K638+193 | K638+199 | 0.767 | IV | 0.188 | I | 0.144 | II |
| 167 | K638+199 | K638+226 | 0.767 | IV | 0.126 | I | 0.097 | II |
| 168 | K638+226 | K638+368 | 0.767 | IV | 0.532 | III | 0.408 | III |
| 169 | K638+368 | K638+409 | 0.767 | IV | 0.604 | III | 0.463 | III |
| 170 | K638+409 | K638+432 | 0.767 | IV | 0.205 | I | 0.157 | II |
| 171 | K638+432 | K638+444 | 0.767 | IV | 0.525 | III | 0.403 | III |
| 172 | K638+444 | K638+676 | 0.767 | IV | 0.173 | I | 0.133 | II |
| 173 | K638+676 | K638+837 | 0.767 | IV | 0.479 | II | 0.367 | III |
| 174 | K639+837 | K639+266 | 0.744 | III | 0.483 | II | 0.359 | III |
| 175 | K639+266 | K639+339 | 0.744 | III | 0.427 | II | 0.318 | III |
| 176 | K639+339 | K639+435 | 0.744 | III | 0.549 | III | 0.408 | III |
| 177 | K639+435 | K639+562 | 0.631 | III | 0.324 | II | 0.204 | II |
| 178 | K640+562 | K640+63 | 0.607 | III | 0.476 | II | 0.289 | III |
| 179 | K641+63 | K641+600 | 0.607 | III | 0.604 | III | 0.367 | III |
| 180 | K642+600 | K642+225 | 0.607 | III | 0.461 | II | 0.280 | III |

612