# Peer review of "Application of the LM-BP neural network approach for"

_Natural Hazards and Earth System Sciences, 2018_

## Referee Comment (RC1) · Anonymous Referee #1 · 25 Dec 2018

The overall logic of this manuscript is clear. However, I don't think this manuscript conveys a lot of valuable information currently. Besides, the description of some research processes is somewhat ambiguous. 1. First of all, it should be pointed out that the neural network method (machine learning) is a hot topic of current research, and it is even expected to become an important force to promote social development and change. Therefore, I am very willing to affirm the author's far-sighted efforts in the field of machine learning. 2. Considering the wide application of neural network methods so far, the novelty and significance of this research need to be articulated. 3. The reason for choosing BP neural network among so many machine learning methods should be articulated. It is suggested making a detailed comparison of different methods. 4. In order to avoid misunderstanding, it should be more appropriate to replace

"hazard assessment" mentioned in the manuscript by "susceptibility assessment", for the meaning of these two expressions is not exactly the same. 5. The expressions like "assessment factor", "evaluation index" and "evaluation indicator" should be consistent. 6. In the "Methods" section, the basic theoretical introduction of BP neural network or entropy weight is not found, which may bring difficulties for readers without relevant foundations to accurately understand the following research. 7. Line 58 The description that "most of these methods" should be specific. 8.Lines 146∼147 Considering that there are many optimization methods for BP neural network, the reason for choosing LM algorithm for optimization should be briefly described. 9. Lines 157∼159 "The classification criteria of the evaluation indicators" in this research need to be articulated, for the solution to this problem is currently inconclusive. 10. Line 181 Correct the "comparison" to "Comparison". 11. Line 187, Line 204 and Line 213 The reason for grading using "the equal interval method" needs to be explained. In fact, the equal interval method may not be the most appropriate choice. 12. Line 284 Correct the "Results and comparison" to "Conclusion". 13. Table 3∼5 The format of the units in the same table should be consistent.

---

## Author Comment (AC1) · 15 Jan 2019

Dear Editor and Referees, First of all, we are very thankful for your constructive comments on our study. Specially, we are heartily grateful to your valuable suggestions. The manuscript has been revised carefully and strictly according to your letter. We are submitting our revised version entitled ""Application of the LM-BP neural network approach for landslide risk assessments", Manuscript ID nhess-2018-360. Please find the revised manuscript with track changes. In order to facilitate your review, bold fonts were used to show revision and changes. In the following "Point-to-point response to the editor's letter and the reviewers' comments". Please do not hesitate to contact me, if further material or information is needed.

[Figure]

Note: All major changes are red-marked in the revised manuscript. Thanks again.

Detailed responses to the comments are addressed below. Reviewers' Comments to Author:

The overall logic of this manuscript is clear. However, I don't think this manuscript conveys a lot of valuable information currently. Besides, the description of some research processes is somewhat ambiguous.

1. First of all, it should be pointed out that the neural network method (machine learning) is a hot topic of current research, and it is even expected to become an important force to promote social development and change. Therefore, I am very willing to affirm the author's far-sighted efforts in the field of machine learning.

Thank you for your comments.

2. Considering the wide application of neural network methods so far, the novelty and significance of this research need to be articulated. Thanks a lot. The first, the system of the vulnerability assessment, considered the pipeline position and the angle between the pipe and the landslide (pipeline laying environmental factors). In previous studies, pipeline vulnerability evaluation indicators only considered the pipeline itself, and the relationship between the pipeline and environment was rarely examined (W. Feng, Zhang, & Zhang, 2014; Shuiping Li, 2008; Yingchun Liu et al., 2015), We also used an interpolation theory to generate the standard sample matrix of the LM-BP neural network. Line 189-194: According to the order of susceptibility from low to high, Interpolation was performed in each interval and the sample vectors of each evaluation indicator were constructed. Each 200 is a susceptibility level, and the sample vector length of each evaluation indicator is 800. The interval of the susceptibility degree is [0, 1], and the output vector is obtained by interpolating 800 values equidistantly between the interval of [0, 1]. Sample matrix is built by interpolation theory, which avoids the excess human influence in the process of building neural network model by traditional methods.

3. The reason for choosing BP neural network among so many machine learning methods should be articulated. It is suggested making a detailed comparison of different methods. Yes, the concrete information of reason for choosing BP neural network has been supplemented. And it make sense to compare different approaches in detail. We will study them in the next step. Line 150-163: BP Neural network with many adjustable parameters has powerful parallel processing mechanism, high flexibility and is good at dealing with a lot of uncertain information. The mechanism of landslide evaluation is complex, with many uncertainties and incomplete information (Jie et al., 2015). The BP neural network model can find out the intrinsic rules from the vast amount of complex and fuzzy data in the changing environment and make corresponding inferences. This method can be applied to the landslide susceptibility assessment of pipeline area with more qualitative information and less quantitative information, and the more accurate assessment results can be obtained from the analysis of these fuzzy information. Landslide susceptibility assessment is essentially a study of pattern recognition (F. Feng, Wu, Niu, Xu, & Yu, 2017). BP neural network can approximate arbitrary continuous function with arbitrary precision, so it is widely used in non-linear modeling, pattern recognition and pattern classification (Xiong, Ran, Xiong, Li, & Ye, 2010). Because the BP neural network model is widely used, there are many successful cases for reference in the number of neurons in each layer, the parameters of network learning and the optimization of algorithms, which can effectively improve the reliability and accuracy of the model(Ke & Li, 2014b).

4. In order to avoid misunderstanding, it should be more appropriate to replace "hazard assessment" mentioned in the manuscript by "susceptibility assessment", for the meaning of these two expressions is not exactly the same.

It is an important question. We thank you for your valuable comments, and all of the "hazard assessment "expressions have been corrected throughout manuscript.

5. The expressions like "assessment factor", "evaluation index" and "evaluation indicator" should be consistent.
Thank you, the expression has been corrected according to the comments of reviewer's

6. In the "Methods" section, the basic theoretical introduction of BP neural network or entropy weight is not found, which may bring difficulties for readers without relevant foundations to accurately understand the following research. Thank you for your valuable comments. The basic theoretical introduction have been added. Line147-150: The neural network, an abstract model of our brain, constructs calculating units connecting with one another. Neural network has an input layer, a hidden layer and an output layer. With its good performance on nonlinear statistical modeling, it is very useful in exploring the hidden relationships between the inputs and the outputs (Z. Wu & Wang, 2016). The flow chart of LM-BP neural network algorithm is shown in Figure 3.

Line 196-208: Entropy is a method of measuring the uncertainty of information by using probability theory (P. Liu & Zhang, 2011). The entropy indicates the extent of difference in an indicator, and the more difference of the data, the greater the role in evaluation (Jia, Zhao, Nan, & Zhao, 2007). The extremum difference method was used to normalize each indicator value. The decision information of each index can be expressed by entropy value ei:

7. Line 58 The description that "most of these methods" should be specific. Thank you, we have re-wrote the sentence. Line 59-62: However, most of these methods are subjective, such as expert evaluations, analytical hierarchy processes, logistic regressions and fuzzy integration methods, which could affect the accuracy and reasonableness of the evaluation (Fall, Azzam, & Noubactep, 2006; Sarkar & Gupta, 2005)

8. Lines 146âĹij147 Considering that there are many optimization methods for BP neural network, the reason for choosing LM algorithm for optimization should be briefly described. Thank you, we have added the reason. Line 166-169: LM algorithm is a combination of gradient descent method and Gauss-Newton method. Its iteration process is no longer along a single negative gradient direction, which greatly improves

the convergence speed and generalization ability of the network (Jing Li, Feng, Wang, & Zhang, 2016).

9. Lines 157âĹij159 "The classification criteria of the evaluation indicators" in this research need to be articulated, for the solution to this problem is currently inconclusive. We thank you for your comments, and classification criteria have been added. Line 180-186: Based on previous research experience and field investigations (Appendix 8), the monotonous intervals of different indicators of susceptibility degrees were judged (Appendix 1). For instance, there were hardly any landslides, only collapses that occurred in slopes above 60 degrees. Besides, the susceptibility degree in the area was monotone decreasing in the interval of [60, 90]. Because of the very small sliding force in a slope at 0 degrees to 15 degrees, landslides were rare to occur here, even under other extreme conditions. (Q. Zhang, Xu, Wu, & Li, 2015)

10. Line 181 Correct the "comparison" to "Comparison". Done

11. Line 187, Line 204 and Line 213 The reason for grading using "the equal interval method" needs to be explained. In fact, the equal interval method may not be the most appropriate choice. Yes, done. We have supplemented the analysis: Line 251-256: Scientific analysis and expression of disaster risk assessment results can simplify complex risk assessment and make the micro results macro (Ding & Tian, 2013). There is no unified criterion for disaster evaluation zoning, and the equal interval method is one of the methods to express the results more intuitively (H. Hu, Dong, & Pan, 2011; Jin & Meng, 2011; Y. Wang, Hao, Zhao, & Fang, 2011). The susceptibility degrees and vulnerability degrees were distinguished using the equal interval method, and four risk grades were then automatically generated.

12. Line 284 Correct the "Results and comparison" to "Conclusion". Done

13. Table 3âĹij5 The format of the units in the same table should be consistent. Yes, done.

[Figure]

Please also note the supplement to this comment:
https://www.nat-hazards-earth-syst-sci-discuss.net/nhess-2018-360/nhess-2018-360-AC1-supplement.zip
* * *
[Figure]

Fig. 1. Figure 3 Flow chart of LM-BP neural network algorithm

---

## Referee Comment (RC2) · Anonymous Referee #1 · 17 Jan 2019

The author had carefully revised the article according to the comments. Acceptance is suggested. Meanwhile, there are still two minor suggestions just for the author's reference. 1. The practical application value and method of this research in engineering geological projects should be better clarified. 2. The future research directions should be indicated.

---

## Author Comment (AC2) · 18 Jan 2019

Dear Editor and Referees, We are heartily grateful to your valuable suggestions. We are submitting our revised version entitled ""Application of the LM-BP neural network approach for landslide risk assessments", Manuscript ID nhess-2018-360. Please find the revised manuscript with track changes. In order to facilitate your review, bold fonts were used to show revision and changes. In the following "Point-to-point response to the editor's letter and the reviewers' comments". Please do not hesitate to contact me, if further material or information is needed.

Note: All major changes are red-marked in the revised manuscript. Thanks again.

Detailed responses to the comments are addressed below. Reviewers' Comments to

[Figure]

Author:

The author had carefully revised the article according to the comments. Acceptance is suggested. Meanwhile, there are still two minor suggestions just for the author's reference.

1. The practical application value and method of this research in engineering geological projects should be better clarified.

Thank you for your comments. Relevant descriptions have been add. Line 332-339ïijŽ The main purpose of this study was to provide managers and planners with a comprehensive assessment of landslide risk in pipeline area. The results offer information on the possibility of failure of slopes or even pipelines in an area in the future, rather than the area that may be damaged by landslides. The landslide susceptibility maps could help planners reorganize and plan future pipeline construction. Pipeline vulnerability maps could assist engineers for pipeline maintenance operation. Based on this final risk map, managers and engineers can then make decisions and formulate prescriptions that will have highly predictable results for safely transporting medium, settlement relocation, and significantly reducing risk of any adverse effects.

2. The future research directions should be indicated.

We thank you for your comments, and we have supplemented relevant descriptions. Line 340-345 Future research could explore detailed comparison of different methods, and finalize an optimal method. Moreover, it is possible that the information needed for the landslide risk assessments can be obtained by simple and effective ways, if plan to build database. Meanwhile, the landslide risk assessment model can be designed as dynamic systems, as the developments in computer and GIS technologies. The system predicts possible future landslides or pipeline damaged by inputting the information obtained in the database, and various adjustment factors.

Please also note the supplement to this comment:

https://www.nat-hazards-earth-syst-sci-discuss.net/nhess-2018-360/nhess-2018-360-AC2-supplement.zip

---

## Referee Comment (RC3) · Anonymous Referee #1 · 25 Jan 2019

The authors had revised the article. Acceptance is suggested.

---

## Referee Comment (RC4) · Anonymous Referee #2 · 30 Jan 2019

I have reviewed both the initial and the revised version of the manuscript 'Application of the LM-BP neural network approach for landslide risk assessments' by Junnan Xiong et al.

The manuscript at its present form emposes a very interesting and good contribution to the application of machine learning techniques in natural hazards research. Moreover, it is clearly visible that the revised manuscript has undergone a considerable improvement compared to the initial version.

Since all of my major objection of the initial manuscript have been clarified in the already revised version, I suggest to consider the revised manuscript for publication.

[Figure]

2018-360, 2018.

---

## Author Comment (AC3) · 14 Feb 2019

We thank you for your constructive criticism that has helped us to improve the manuscript.

---

## Author Comment (AC4) · 14 Feb 2019

Thanks very much for your kind work and affirmation on our paper. On behalf of my co-authors, we would like to express our great appreciation to you.
* * *

---

## Author Response (AR1)

Dr. Junnan Xiong

School of Civil Engineering and Architecture

Southwest Petroleum University

Chengdu 610500

China

Phone: +86 13541223403

neu_xjn@163.com

February 26, 2019

Subject: Manuscript revision

Dear Editor,

First of all, we are very thankful for your constructive comments on our study. Specially, we are

heartily grateful to your valuable suggestions.

The manuscript has been revised carefully and strictly according to your letter. We are submitting

our revised version entitled ""Application of the LM-BP neural network approach for landslide risk

assessments", Manuscript ID nhess-2018-360.

Please find the revised manuscript with track changes. In order to facilitate your review, bold fonts

were used to show revision and changes. In the following "Point-to-point response to the editor's letter

and the reviewers' comments".

Please do not hesitate to contact me, if further material or information is needed.

Note: All major changes are red-marked in the revised manuscript.

Thanks again.

Sincerely yours,

Junnan Xiong

Detailed responses to the comments are addressed below.

Comments to the Author:

I have read with interest your manuscript and I"m pleased to see the effort you have taken to include earlier comments. However, I think the manuscript can improve when you consider the below comments.

• Reword running title to be more specific. For example reword to 'Landslide risk zonation for areas containing hydrocarbon transport pipelines'.

**Thank you for your comments. We have reworded the running title.**

**Running Title: Landslide risk zonation for areas containing Products oil transport pipelines**

• Line 35&36. Change the word 'mileage' to 'length'. Mileage has implicitly the meaning of a none SI unit 'mile'.

**Yes, done**

• Line 119 – 121. 'The irrational unit …..'. Explain what 'the irrational unit' is.

**Thank you, we have added the explanation.**

**Line 122:**

**The irrational unit (slope unit with inaccurate boundary) was artificially identified and modified by comparing GF-1 satellite remote sensing images. Boundary correction, fragment combination and fissure filling were used for modification.**

• Line 122. Reword 'The object of the pipeline vulnerability assessment in the landslide area was the pipeline' to something like: 'This study focusses on assessing the vulnerability of transport pipelines to landslides'.

**Done**

• Line 131. Remove '…of the paper…'

**Done**

• Line 147-148. Replace 'The neural network, an ….. and an output layer.' with 'A neural network is a nonlinear mathematical structure which is capable of representing complex nonlinear processes that relate the inputs and outputs of any system (Hsu et al., 1995).'

**Yes, done**

• Line 151-152. Replace '…and is good at dealing with a lot of uncertainty information.' With '…and can incorporate well uncertainty information.'

**Done**

• Line 155-157. Reword the sentence 'This method can be … these fuzzy information.' I'm unsure what you try to say here.

We thank you for your comments, and the sentence have been reword.

**Line: 159-161**

**The information about landslide reflected by the data used in the process of susceptibility assessment is mostly qualitative rather than quantitative. Through the analysis of these fuzzy information, accurate assessment results can be obtained.**

• Line 183-184. Double check your interval and reword such that it reads something like: 'Besides, the susceptibility degree in the area was monotone decreasing in the slope interval of 60 to 90 degrees.' So do you mean the slope interva l of 60 to 90 degrees?

**Yes, done. It means the slope interva l of 60 to 90 degrees**

• Line 184. Change '…in a slope at ….' To '…in slopes at…'.

**Done**

• Line 190-193. Reword 'Each 200 is… by traditional methods.' It is unclear what you mean here.

**Thank you, we have rewritten the sentence.**

**Line 193-199:**

**When establishing the empty matrix, the sample size of each landslide susceptibility level was set to 200, and the training sample size was 800. According to the order of susceptibility from low to high (Appendix 1), the input was constructed by interpolating in each interval. The interval of the susceptibility degree was [0, 1], and the output was obtained by interpolating 800 values equidistantly between the interval of [0, 1] (Appendix 2). Using interpolation theory to build samples avoided the excess human influence in the process of building neural network model by traditional methods.**

• Line 204. Reword '…indicator, and the more difference of the data, the…' to something like

'…indicator, the more difference the data, the…'.

**Done**

• Line 258. Reword '…and make the micro results macro.' To something like '…and accelerate

findings'.

**Done**

• Line 261. Remove 'degrees' such that it reads: 'The susceptibility and vulnerability degrees were

distinguished ….'.

**Yes, done**

• Line 332. Remove 'with' such that it reads: '…and planners a comprehensive…'.

**Yes, done**

• Line 333. Change such that it reads: '…of landslide risk in areas containing pipelines.'.

**Done**

• Line 334. Remove '… or even pipelines in an area….. by landslides.' Such that it reads: '… of failure

of slopes.'

**Done**

• Line 340. Reword such that it reads: '…of different methods and recommend one or more optimal approaches.'.

**Done**

• Line 342-345. Reword to something like: 'This study shows that landslide risk assessments can be performed with minimal amount of relatively easy to obtain datasets. We advocate to establish a database with assessment parameters similar as described by this study to construct dynamic landslide risk assessment models.'

**Yes, done**